Expression and prognostic analyses of ITGA11, ITGB4 and ITGB8 in human non-small cell lung cancer

Wu Pancheng 1
Wang Yanyu 1
Wu Yijun 2
Jia Ziqi 2
Song Yang 3
Liang Naixin pumchnelson@163.com 3
1 Department of Thoracic Surgery, Peking Union Medical College Hospital, Peking Union Medical College, Chinese Academy of Medical Sciences , Beijing , China
2 Peking Union Medical College, Eight-Year MD Program, Chinese Academy of Medical Sciences , Beijing , China
3 Department of Thoracic Surgery, Peking Union Medical College Hospital, Chinese Academy of Medical Sciences , Beijing , China
Nakai Kenta
Electronic publication date: 2019 Dec 20
Publication date: 2019
Volume: 7
Electronic Location ID: e8299
Received 2019 Aug 20; Accepted 2019 Nov 26
Copyright: ©2019 Wu et al.
Copyright year: 2019
Copyright holder: Wu et al.
License: This is an open access article distributed under the terms of the Creative Commons Attribution License, which permits unrestricted use, distribution, reproduction and adaptation in any medium and for any purpose provided that it is properly attributed. For attribution, the original author(s), title, publication source (PeerJ) and either DOI or URL of the article must be cited.
License URL: https://creativecommons.org/licenses/by/4.0/

Keywords: ITGA11, ITGB4, ITGB8, Non-small cell lung cancer, Expression and prognostic analysis

Funding: Natural Science Foundation of Beijing, China 7182132 This work was supported by the Natural Science Foundation of Beijing, China (Grant No. 7182132). The funders had no role in study design, data collection and analysis, decision to publish, or preparation of the manuscript.

==============================
Background

Integrins play a crucial role in the regulation process of cell proliferation, migration, differentiation, tumor invasion and metastasis. ITGA11, ITGB4 and ITGB8 are three encoding genes of integrins family. Accumulative evidences have proved that abnormal expression of ITGA11, ITGB4 and ITGB8 are a common phenomenon in different malignances. However, their expression patterns and prognostic roles for patients with non-small cell lung cancer (NSCLC) have not been completely illustrated.

Methods

We investigated the expression patterns and prognostic values of ITGA11, ITGB4 and ITGB8 in patients with NSCLC through using a series of databases and various datasets, including ONCOMINE, GEPIA, HPA, TCGA and GEO datasets.

Results

We found that the expression levels of ITGA11 and ITGB4 were significantly upregulated in both LUAD and LUSC, while ITGB8 was obviously upregulated in LUSC. Additionally, higher expression level of ITGB4 revealed a worse OS in LUAD.

Conclusion

Our findings suggested that ITGA11 and ITGB4 might have the potential ability to act as diagnostic biomarkers for both LUAD and LUSC, while ITGB8 might serve as diagnostic biomarker for LUSC. Furthermore, ITGB4 could serve as a potential prognostic biomarker for LUAD.

Introduction

Lung cancer is the most frequent malignancy and the leading cause of cancer-related death all over the world. Five-year survival rate for lung cancer patients ranges from 4% to 17% depending on disease stage and regional differences (Hirsch et al., 2017). Non-small cell lung cancer (NSCLC) is the most common pathological type of lung cancer and responsible for 85% to 90% of all lung cancer (Osmani et al., 2018). Owing to the problems in early diagnosis, patients with NSCLC are often diagnosed at advanced stage, which contributes a lot to the dismal prognosis (Ellis & Vandermeer, 2011; Jan et al., 2019). Thus, there is an urgent need to discover new diagnostic and prognostic biomarkers for NSCLC.

Integrins function as bridges between the extracellular matrix (ECM) and the cytoskeleton and work as radars to detect changes in the cellular microenvironment, which enables cells to react according the external milieu (Bianconi, Unseld & Prager, 2016; Ginsberg, 2014). They play a crucial role in the regulation process of cell proliferation, migration, differentiation, tumor invasion and metastasis (Slack-Davis & Parsons, 2004). Integrins family include 24 different transmembrane, multifunctional heterodimers and are composed of an α and a β subunit (Brakebusch et al., 2002). There are 18 different α subunits and eight different β subunits in human body (Hynes, 1992). Recently, the effects of integrins in tumor progression have been receiving a great deal of attention.

ITGA11 encodes integrin subunit α11, which dimerizes with β1 subunit and forms as a cell surface collagen receptor involved in the process of cell migration and collagen reorganization (Tiger et al., 2001). Integrin α11 was overexpressed in the stroma of most head and neck squamous cell carcinomas (HNSCC) and correlated positively with alpha smooth muscle actin expression (Parajuli et al., 2017). In addition, ITGA11 was overexpressed by cancer-associated fibroblast (CAFs) in Pancreatic Ductal Adenocarcinoma (PDAC) stroma and may serve as an interesting stromal therapeutic target (Schnittert et al., 2019). Integrin subunit β4, also known as a laminin-5 receptor, is a protein encoded by ITGB4 (Wang et al., 2012). Inhibition of ITGB4 in glioma cells would decrease the self-renewal abilities of glioma stem cells and suppress the malignant behaviors of glioma cells in vitro and in vivo (Ma et al., 2019). Moreover, higher ITGB4 expression level was detected in tumor than adjacent non-tumor tissues in patients with hepatocellular carcinoma (HCC). Silencing of ITGB4 could repress cell proliferation, colony forming ability and cell invasiveness (Li et al., 2017). Integrin β8, paired with αv subunit, is encoded by ITGB8. It has been reported that ITGB8 is upregulated in laryngeal squamous cell carcinoma (Ni et al., 2012). Additionally, the expression level of ITGB8 can be regulated by the tumor-promoting receptor tyrosine kinase-EphB4, while knockdown of ITGB8 may suppress migration and invasion in prostate cancer cell lines (Mertens-Walker et al., 2015). These studies have shown that ITGA11, ITGB4 and ITGB8 might be candidate biomarkers and therapeutic targets with great potential.

Recent years, there have been developed multifarious platforms, databases as well as various datasets on the web that allow cancer researchers to make in-depth bioinformatic analysis in cancer with multi omics data. Several prognostic biomarkers with great potential for NSCLC have also been identified. For instance, it has been reported that STMN1 expression was correlated with poor OS in patients with Squamous Cell Lung Carcinoma (LUSC) and might serve as a prognostic biomarker (Bao et al., 2017). Using bioinformatics methods, Xie et al. (2019) have found that KRT8 expression might be an independent prognostic biomarker for poor OS and PFS in Lung Adenocarcinoma (LUAD). Sun et al. (2019) have identified five genes that could predict metastasis in NSCLC and might serve as potential targets. As far as we know, bioinformatics analysis has not been applied to explore the roles of ITGA11, ITGB4 and ITGB8 in NSCLC. Therefore, we conducted this study to analyze the expression patterns and prognostic values of these three genes in NSCLC based on online databases, platforms and various datasets.

Materials and Methods

ONCOMINE analysis

The expression levels of ITGA11, ITGB4 and ITGB8 and genes co-expressed with ITGA11, ITGB4 and ITGB8 were analyzed in ONCOMINE database (https://www.oncomine.org) (Rhodes et al., 2007; Rhodes et al., 2004). The cut-off of p value and fold change were defined as 0.01 and 2, respectively (Huang et al., 2019).

GEPIA (Gene Expression Profiling Interactive Analysis) analysis

GEPIA (http://gepia.cancer-pku.cn/) is an interactive web application for gene expression analysis based on 9736 tumors and 8587 normal samples from the TCGA (The Cancer Genome Atlas) and the GTEx (Genotype-Tissue Expression) databases (Tang et al., 2017). The GEPIA database was used to compare mRNA levels of ITGA11, ITGB4 and ITGB8 between TCGA and GTEx databases. Meanwhile, the association among ITGA11, ITGB4 and ITGB8 in NSCLC were also analyzed in GEPIA.

Bioinformatics analysis of data using The Cancer Genome Atlas lung cancer datasets

The level 3 data of TCGA-LUAD and TCGA-LUSC were obtained from UCSC Xena platform (https://xenabrowser.net/datapages/) (Goldman et al., 2015) and RTCGA package (https://rtcga.github.io/RTCGA). The LUAD and LUSC gene expression RNAseq datasets included 524 tumor tissues and 499 tumor tissues, respectively. 502 of the LUAD patients and 492 of the 499 LUSC patients had complete survival data. The differences in overall survival (OS) of LUAD and LUSC patients with high and low expression of ITGA11, ITGB4 and ITGB8 were assessed by Kaplan–Meier curves. Meanwhile, the association between tumor stage and the expression levels of ITGA11, ITGB4 and ITGB8 were also analyzed. Clinicopathological parameters, including age at diagnosis, gender, vital status, tumor stage, smoking history and OS time, were extracted for univariate and multivariate cox regression analysis.

Gene Expression Omnibus (GEO) microarray datasets analysis

To validate the expression profiles of ITGA11, ITGB4 and ITGB8 in NSCLC, we collected a total of 21 datasets including tumor and non-tumor tissues of NSCLC in GEO database (https://www.ncbi.nlm.nih.gov/geo/). We analyzed the mRNA levels of ITGA11, ITGB4 and ITGB8 between tumor and non-tumor controls for each GEO dataset. In addition, we performed a meta-analysis based on the enrolled GEO microarray datasets.

Immunohistochemistry analysis

The protein expression of ITGA11, ITGB4 and ITGB8 in normal lung and tumor tissues were examined using the Human Protein Atlas (HPA) (https://www.proteinatlas.org/) (Uhlen et al., 2015; Uhlen et al., 2017).

Statistical analysis

Statistical analysis was performed on R software (3.6.1) (https://www.r-project.org/) and an integrated development environment RStudio (1.2.1335) (https://rstudio.com/). The mRNA expression of ITGA11, ITGB4 and ITGB8 between NSCLC tissues and normal controls were compared using Student’s t-test. Data visualization was performed using an R package called “ggstatsplot” (https://CRAN.R-project.org/package=ggstatsplot). Kaplan–Meier curves of OS were performed in TCGA-LUAD and TCGA-LUSC raw data by setting median expression of ITGA11, ITGB4 and ITGB8 as cut-off. Statistical differences were assessed by the log-rank test. Univariate and multivariate survival analyses were performed using cox regression model, risk factors (p < 0.2) analyzed by univariate analysis were selected for multivariate analysis.

For GEO datasets analysis, mean (M) and standard deviation (SD) were calculated for each NSCLC tumor and normal control group. In addition, an R package called “meta” was used in R to perform a comprehensive meta-analysis (Schwarzer, 2007). The Q test and I2 statistic were calculated to assess the heterogeneity among the enrolled studies. If p < 0.05 or I2>50%, a random effects model would be selected. Sensitivity analysis was conducted to explore whether a specific study played a crucial influence in significant heterogeneity. Finally, the publication bias was examined through funnel plots and Egger’s test (Egger et al., 1997). Once there was a publication bias, the “fill and trim” method would be selected to adjust for the bias (Duval & Tweedie, 2000). p < 0.05 deemed statistically significant.

Results

The expression levels of ITGA11,ITGB4andITGB8 in patients with non-small cell lung cancer

Using ONCOMINE database, we investigated the transcription levels of ITGA11, ITGB4 and ITGB8 in lung cancer vs. normal samples. ONCOMINE analysis revealed that the mRNA expression of ITGA11, ITGB4 and ITGB8 were obviously overexpressed in NSCLC tissues in ten datasets (Fig. 1). These datasets were summarized in Table 1. The GEPIA analysis results also suggested that the expression levels of ITGA11 and ITGB4 were significantly higher in both LUAD and LUSC than that in normal tissues, while the expression level of ITGB8 was only significantly upregulated in LUSC tissues (Fig. 2). Furthermore, we analyzed ITGA11, ITGB4 and ITGB8 mRNA expression level in both lung cancer and normal tissues using the TCGA-LUAD and TCGA-LUSC original data. The results revealed that the expression levels of ITGA11, ITGB4 and ITGB8 were all significantly upregulated in tumor tissues compared with normal tissues (Fig. S1).

Figure 1 The transcription levels of ITGA11, ITGB4 and ITGB8 in different cancers compared with normal tissues in the ONCOMINE dabase.

Cell color is determined by the best gene rank percentile for the analysis within the cell.

Table 1 The transcription levels of ITGA11, ITGB4 and ITGB8 between lung cancer and normal samples in ONCOMINE database.

Gene ID	Types of lung cancer vs. normal	Fold change	P value	t-Test	References	
ITGA11	Lung Adenocarcinoma vs. Normal	2.047	6.79E–16	10.685	Selamat et al. (2012)	
Lung Adenocarcinoma vs. Normal	2.968	7.47E–09	7.945	Okayama et al (2012)	
ITGB4	Squamous Cell Lung Carcinoma vs. Normal	2.867	1.32E–05	8.706	Wachi et al. (2005)	
Squamous Cell Lung Carcinoma vs. Normal	3.505	5.33E–06	6.406	Garber et al. (2001)	
Squamous Cell Lung Carcinoma vs. Normal	2.637	4.64E–10	7.458	Talbot et al. (2005)	
Squamous Cell Lung Carcinoma vs. Normal	6.818	5.21E–04	3.57	Bhattacharjee et al. (2001)	
Lung Adenocarcinoma vs. Normal	2.99	1.17E–14	9.575	Selamat et al. (2012)	
Squamous Cell Lung Carcinoma vs. Normal	3.591	8.92E–10	8.599	Hou et al. (2010)	
ITGB8	Squamous Cell Lung Carcinoma vs. Normal	2.455	1.95E–05	5.627	Garber et al. (2001)	
Squamous Cell Lung Carcinoma vs. Normal	2.876	1.26E–07	6.484	Hou et al. (2010)	

Figure 2 The expression levels of ITGA11 (A), ITGB4 (B) and ITGB8 (C) between NSCLC tissues and normal tissues in GEPIA.

*Indicate that the results are statistically significant.

To further explore the protein expression of ITGA11, ITGB4 and ITGB8 in NSCLC, we analyzed the IHC images using the Human Protein Atlas (HPA) database. As shown in Fig. 3, the protein expression of ITGA11 and ITGB4 were upregulated in both LUAD and LUSC cancer tissues compared with normal lung tissues (Figs. 3A–3C and 3D–3F). In comparison, the protein expression of ITGB8 was obviously upregulated in LUSC with medium staining, but not in LUAD (Figs. 3G–3I).

Figure 3 Immunohistochemistry analysis for ITGA11, ITGB4 and ITGB8 in NSCLC (HPA database).

(A–F) The protein expression of ITGA11 and ITGB4 were significantly higher in both LUAD and LUSC tissues compared with the normal lung, respectively. (G–I) The protein expression level of ITGB8 was significantly higher in LUSC tissues compared with the normal lung.

Confirmation of the expression profiles of ITGA11,ITGB4andITGB8 in non-small cell lung cancer using GEO datasets

We also performed a data-mining analysis to investigate the differences in the expression levels of ITGA11, ITGB4 and ITGB8 between tumor and normal tissues in NSCLC using GEO datasets. The main characteristics of the enrolled GEO studies were described in Table S1. The results were shown in Fig. 4 and Figs. S2–S4. As illustrated in Fig. 4A and Fig. S2D, the expression level of ITGB4 was significantly increased in tissues from patients with LUAD (SMD: 0.94; 95% CI [0.65–1.24]; p < 0.01) as well as LUSC (SMD:1.37; 95% CI [0.71–2.04]; p < 0.01) compared to the normal tissues. The heterogeneity was apparent for LUAD (I 2= 80%; p < 0.01) and LUSC (I 2= 89%; p < 0.01). The following sensitivity analysis demonstrated that no study was found to have a vital influence in the enrolled studies (Fig. 4B and Figs. S3D). In addition, we didn’t find evidence of publication bias based on the funnel plot and the Egger’s test (Fig. 4C, p = 0.7759). However, the Figs. S4D indicated publication bias (Egger’s test, p = 0.04729). Therefore, we used the fill and trim method to adjust for the bias. The adjusted random effects model result showed that ITGB4 was also significantly upregulated in LUSC tissues (SMD: 0.77; 95% CI [0.03–1.52]; p = 0.04).

Figure 4 Meta-analysis of ITGB4 expression in LUAD tissues compared with normal controls based on GEO datasets.

(A) Forest plot of SMD comparing ITGB4 expression in LUAD tissues with normal controls from the enrolled GEO datasets. (B) Sensitivity analysis of the enrolled GEO datasets. (C) The evaluation of the publication bias of the enrolled GEO datasets (Egger’s test, p = 0.7759).

The analysis results of ITGA11 and ITGB8 mRNA levels in LUAD and LUSC were the same as the above results (Figs. S2–S4). The separate analyses of the expression levels of ITGA11, ITGB4 and ITGB8 in LUAD and LUSC tissues compared with normal tissues for each GEO dataset were presented in the Figs. S5 and S6.

The prognostic values of ITGA11,ITGB4andITGB8 in non-small cell lung cancer

By using GEPIA, we investigated the prognostic values of ITGA11, ITGB4 and ITGB8 in NSCLC. The survival curves revealed that high expression level of ITGB4 could indicate a poor OS in LUAD (p <0.001; Fig. 5B), while ITGA11 and ITGB8 were not related with OS in LUAD (p = 0.064 and p = 0.78, respectively, Figs. 5A and 5C). In comparison, there were no obvious associations between the expression levels of ITAG11, ITGB4 and ITGB8 and LUSC (Figs. 5D–5F). Moreover, using the TCGA original data, we performed survival analysis to validate these associations. The results were consistent with GEPIA analysis (Fig. S7).

Figure 5 Kaplan–Meier survival curves of overall survival (OS) in LUAD and LUSC (GEPIA database).

Survival curves of OS based on the high and low expression of ITGA11, ITGB4 and ITGB8 in LUAD (A–C) and LUSC (D–F), respectively.

Next, we performed cox regression analysis to further assess and validate the prognostic values of ITGA11, ITGB4 and ITGB8 in NSCLC based on TCGA original data. The univariate cox analysis indicated that high ITGB4 expression and advanced stages were significantly correlated with worse OS in LUAD (Table 2). Meanwhile, multivariate cox analysis confirmed that high ITGB4 expression was an independent prognostic biomarker for patients with LUAD (HR: 1.417; 95%CI [1.042–1.926]; p = 0.026; Table 2). In addition, no significant results were found with other genes in the OS of LUAD and LUSC (Table 2). These results were consistent with that analyzed by GEPIA. Furthermore, we investigated the correlation between tumor stage and the expression levels of ITGA11, ITGB4 and ITGB8 (Fig. S8). The results showed that there was a significant correlation between tumor stage and mRNA expression of ITGB8 in LUSC (Fig. S8F).

Table 2 Univariate and multivariate cox analysis of OS in LUAD and LUSC.

Smoking history: 1. lifelong non-smoker; 2. current smoker; 3. current reformed smoker (for >15 years); 4. Current reformed smoker (for ≤ 15 years); 5. current reformed smoker (duration not specified).

Characteristics	Univariate analysis	Multivariate analysis	
	pvalue	HR	95%CI	pvalue	HR	95%CI	
LUAD-OS	
Gender Male vs. Female	0.745	1.050	0.784-1.405	
Age
>65 vs. ≤65	0.229	1.198	0.892-1.610	
Smoking history 2∕3∕4∕5 vs. 1	0.530	0.875	0.578-1.325	
Clinical stage III/IV vs. I/II	0	2.466	1.786–3.404	0	2.329	1.682–3.226	
ITGA11 expression
High vs. Low	0.076	1.306	0.973–1.753	0.361	1.153	0.849–1.566	
ITGB4 expression
High vs. Low	0.002	1.575	1.175–2.112	0.026	1.417	1.042–1.926	
ITGB8 expression
High vs. Low	0.925	0.986	0.737–1.320	
LUSC-OS	
Gender Male vs. Female	0.179	1.251	0.902–1.736	0.177	1.253	0.903–1.739	
Age
>65 vs. ≤65	0.124	1.253	0.940–1.670	0.049	1.343	1.001–1.803	
Smoking history 2∕3∕4∕5 vs. 1	0.430	0.698	0.286–1.704	
Clinical stage III/IV vs. I/II	0.002	1.655	1.199–2.284	0.002	1.665	1.204–2.301	
ITGA11 expression
High vs. Low	0.385	1.128	0.860–1.479				
ITGB4 expression
High vs. Low	0.388	1.127	0.859–1.479				
ITGB8 expression
High vs. Low	0.875	0.978	0.746–1.283				

Co-expression and correlation analyses of ITGA11,ITGB4andITGB8 in non-small cell lung cancer

The co-expression analysis was conducted using ONCOMINE database. Based on Hou Lung dataset (Hou et al., 2010), we analyzed genes that were co-expressed with ITGA11, the result showed that ITGA11 was co-expressed with COL10A1, THBS2, SULF1, CTRHC1, GREM1, C5orf46, COL11A1, NOX4 (Fig. S9A). The Bild Lung dataset indicated that ITGB4 was co-expressed with LAD1, SFN, FXYD3, KRT19, DSG2, JUP, DSP, PERP (Bild et al., 2006) (Fig. S9B). Based on Yamagata Lung dataset (Yamagata et al., 2003), we analyzed genes that were co-expressed with ITGB8, the result showed that ITGB8 was co-expressed with ERC2, PDE6D, C17orf99, SNRNP27, C1orf61, GATA1, PPP2R2B, CCK, CRYBA1, APBA3, CYP3A4, UROS (Fig. S9C).

By using GEPIA, we investigated the association among ITGA11, ITGB4 and ITGB8 in NSCLC based on Pearson correlation analysis. The results indicated that there was no correlation between ITGA11 and ITGB4 (R =  − 0.018; p > 0.05) (Fig. S10A). Also, there was scarcely any correlation between ITGA11 and ITGB8 (R = 0.069; p <  0.05) (Fig. S10B). In addition, a weak positive correlation was found between ITGB8 and ITGB4 (R = 0.32; p  <  0.05) (Fig. S10C).

Discussion

Numerous studies have suggested that ITGA11, ITGB4 and ITGB8 are involved in migration, epithelial-mesenchymal transition, invasion, and metastasis in different cancers (Gan et al., 2018; Huang et al., 2017; Kitajiri et al., 2002; Li et al., 2017). The aberrant expression of ITGA11, ITGB4, and ITGB8 have been reported in many cancers (Grossman et al., 2000; Mertens-Walker et al., 2015; Parajuli et al., 2017; Tagliabue et al., 1998). Regrettably, the expression profiles and prognostic roles of ITGA11, ITGB4 and ITGB8 in NSCLC are still not clear. Thus, we conducted this study to explore the expression patterns and prognostic values of ITGA11, ITGB4 and ITGB8 in NSCLC.

It has been reported that ITGA11 could serve as an important stromal factor in NSCLC, which can enhance tumorigenicity of human non-small cell lung cancer cells by regulating IGF2 expression in fibroblasts (Zhu et al., 2007). Moreover, in carcinoma-associated fibroblasts (CAFs), ITGA11 signaling pathway may play an important role in carcinoma-associated fibroblasts (CAFs), which means Integrin α11β1 can promote tumor growth and metastatic potential of NSCLC cells by regulating cancer stromal stiffness (Navab et al., 2016). These results suggested that ITGA11 might play an important role for NSCLC. In our study, ONCOMINE analysis showed that mRNA expression level of ITGA11 was highly expressed in Lung Adenocarcinoma compared with that in normal controls. GEPIA revealed that the expression level of ITGA11 was obviously higher in both LUAD and LUSC than that in normal tissues. In addition, we also downloaded TCGA original data, GEO datasets, and protein expression data from HPA to validate ITGA11 expression profile, the results were consistent with the GEPIA analysis results. These results indicated that ITGA11 might be a diagnostic biomarker for patients with LUAD and LUSC. Furthermore, we investigated the association between the expression level of ITGA11 and OS in LUAD and LUSC using GEPIA and cox regression analysis. However, the results showed ITGA11 expression had no prognostic role in terms of OS in LUAD and LUSC.

ITGB4 was found to have a strong positive correlation with tumor size (p = 0.01) and tumor nuclear grade (p < 0.01) in early breast cancer (Diaz et al., 2005). Furthermore, it is reported that ITGB4 could promote the invasion and metastasis of tumor cells through a series of processes (Stewart & O’Connor, 2015). These results imply us that ITGB4 might also play a crucial role in NSCLC. In our report, ONCOMINE and GEPIA analysis revealed that the expression level of ITGB4 was significantly upregulated in LUAD and LUSC. Additionally, we confirmed this expression feature by analysis TCGA original data and GEO datasets. The protein level was also consistent with the mRNA expression level. Taken together, these results implied that ITGB4 expression could act as a diagnostic biomarker for patients with LUAD and LUSC. Moreover, the survival curve showed that high ITGB4 expression was strong correlated with inferior OS in LUAD. The following univariate cox and multivariate cox regression analysis confirmed that high ITGB4 expression level was an independent prognostic biomarker for poor OS in LUAD.

It has been reported that ITGB8 could mediate the activation of latent TGF- β, which subsequently derives the epithelial-to-mesenchymal (EMT) transition of some cancers and contributes to cancer cell migration and growth (Mu et al., 2002; Pozzi & Zent, 2011). Furthermore, ITGB8 was significantly upregulated in ovarian cancer tissues compared with that in normal ovary tissues (He et al., 2018). Moreover, It has been reported that ITGB8 silencing could suppress the metastatic potential of human lung cancer cell lines A549 and PC (Xu & Wu, 2012). These studies suggested that ITGB8 might play an important role in NSCLC. In our study, we found that the mRNA expression level of ITGB8 was highly overexpressed in LUSC both in ONCOMINE and GEPIA analysis. This expression feature was successfully validated by analyzing the TCGA original data and GEO datasets. These results suggested that ITGB8 might act as a diagnostic biomarker in LUSC. It was worth mentioning that there was no significant correlation in ITGB8 expression level between LUAD and normal tissues by GEPIA analysis. However, the expression feature was not showed when we analyzed the TCGA original data and GEO datasets. This may due to the lack of normal controls in TCGA datasets and the differences in enrolled participants in GEO datasets. Future large-scale studies are required to assess and validate this expression pattern. In addition, we explored the association between the expression level of ITGB8 and OS in LUAD and LUSC using GEPIA and cox regression analysis. the results showed ITGB8 expression had no prognostic role in terms of OS in LUAD and LUSC. Furthermore, we found that there was a strong correlation between ITGB8 expression level and tumor stage in LUSC.

The potential limitations of our study need to be noted. First, the biological mechanisms of these three candidate markers in LUAD and LUSC are still unknown. Second, although this study had a comprehensive analysis based on several databases such as TCGA and GEO, traditional in-house experimental studies including enough specimens are required to further validate our findings.

Conclusions

In summary, we systematically analyzed the expression patterns and prognostic values of ITGA11, ITGB4 and ITGB8 in patients with LUAD and LUSC by conducting a bioinformatics analysis based on several web platforms and various datasets. Our results indicated that ITGA11 and ITGB4 might act as diagnostic biomarkers for both LUAD and LUSC, while ITGB8 may serve as diagnostic biomarker for LUSC. Furthermore, ITGB4 might serve as a potential prognostic biomarker for LUAD. We hope our findings will enrich the knowledge of diagnostic and therapy designs for patients with NSCLC.

Supplemental Information

Figure S1 (A–C) The expression levels of ITGA11, ITGB4 and ITGB8 between LUAD and normal tissues. (D–F) The expression levels of ITGA11, ITGB4 and ITGB8 between LUSC and normal tissues

Click here for additional data file.

Figure S2 The forest plots for the enrolled GEO datasets

(A–B) The forest plots of overall analysis of ITGA11 and ITGB8 between LUAD patients and normal controls, respectively. (C–E) The forest plots of overall analysis of ITGA11, ITGB4 and ITGB8 between LUSC patients and normal controls, respectively.

Click here for additional data file.

Figure S3 Sensitivity analysis for the enrolled GEO datasets

(A–B) Sensitivity analysis for the enrolled GEO datasets in analyzing ITGA11 and ITGB8 expression between LUAD patients and normal controls, respectively. (C–E) Sensitivity analysis for the enrolled GEO datasets in analyzing ITGA11, ITGB4 and ITGB8 expression between LUSC patients and normal controls, respectively.

Click here for additional data file.

Figure S4 Funnel plots and Egger’s test for the enrolled GEO datasets

(A–B) Funnel plots and Egger’s test for the enrolled GEO datasets in analyzing ITGA11 and ITGB8 expression between LUAD tissues and normal controls, respectively. (C–E) Funnel plots and Egger’s test for the enrolled GEO datasets in analyzing ITGA11, ITGB4 and ITGB8 expression between LUSC tissues and normal controls, respectively.

Click here for additional data file.

Figure S5 Box plots displaying the expression levels of ITGA11, ITGB4 and ITGB8 between LUAD and normal tissues for each GEO dataset

(A–C) The expression levels of ITGA11, ITGB4 and ITGB8 between LUAD and normal tissues for each GEO dataset, respectively.

Click here for additional data file.

Figure S6 Box plots displaying the expression levels of ITGA11, ITGB4 and ITGB8 between LUSC and normal tissues for each GEO dataset

(A–C) The expression levels of ITGA11, ITGB4 and ITGB8 between LUSC and normal tissues for each GEO dataset, respectively.

Click here for additional data file.

Figure S7 Kaplan-Meier curves of overall survival (OS) in LUAD and LUSC patients based on TCGA original data

Survival curves of OS based on the high and low expression of ITGA11, ITGB4 and ITGB8 in LUAD (A–C) and LUSC (D–F), respectively.

Click here for additional data file.

Figure S8 The association between the expression levels of ITGA11, ITGB4 and ITGB8 and tumor stages based on TCGA original data

The association between the expression levels of ITGA11, ITGB4 and ITGB8 and tumor stages in LUAD (A–C) and LUSC (D–F), respectively.

Click here for additional data file.

Figure S9 (A–C) The co-expression analysis result of ITGA11, ITGB4 and ITGB8, respectively

Click here for additional data file.

Figure S10 (A) The relationship between ITGA11 and ITGB4. (B) The relationship between ITGA11 and ITGB8. (C) The relationship between ITGB8 and ITGB4

Click here for additional data file.

Table S1 The main characteristics of the enrolled GEO studies

Click here for additional data file.

Supplemental Information 1 Raw code

Click here for additional data file.

We’d like to thank Dr. Yi Zheng for her kind encouragement during the study period.

Additional Information and Declarations

Competing Interests

Author Contributions

Data Availability

The authors declare there are no competing interests.

Pancheng Wu and Yanyu Wang conceived and designed the experiments, performed the experiments, analyzed the data, authored or reviewed drafts of the paper, approved the final draft.

Yijun Wu, Ziqi Jia and Yang Song analyzed the data, prepared figures and/or tables, approved the final draft.

Naixin Liang conceived and designed the experiments, authored or reviewed drafts of the paper, approved the final draft.

The following information was supplied regarding data availability:

The TCGA-LUAD and TCGA-LUSC raw data is available at UCSC Zena (available at https://xenabrowser.net/datapages/).

NCBI GEO datasets: GSE32863, GSE63459, GSE75037, GSE43458, GSE10072, GSE31547, GSE7670, GSE46539, GSE27262, GSE18842, GSE21933, GSE31552, GSE74706, GSE19188, GSE118370, GSE134381, GSE103512, GSE2088, GSE12428, GSE33479 and GSE30219.

The raw code is available in a Supplementary Files.

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
