# Peer review of "Expression and prognostic analyses of ITGA11, ITGB4 and ITGB8 in human non-small cell lung cancer"

_PeerJ, doi:10.7717/peerj.8299_

## Round 0.1 · original submission · Major Revisions

Your manuscript has been reviewed by three experts in the field. As you can see from their comments below, all of them basically admit the value of this work while raising a number of points. Particularly, two of them ask why the authors focus on the three genes in the integrin family. In addition, more consistent validation methods between them would be desirable. Please read their comments carefully and revise the manuscript accordingly. Note also that some of the reviewers request its English editing.

·

Basic reporting

1. This manuscript discusses the potential ability of ITGA11, ITGB8 and ITGB4 to act as diagnostic and prognostic biomarker for patients with NSCLC (LUAD + LUSC) and LUSC by using several online web sever including ONCOMINE, GEPIA, CCLE, EMBL-EBI, and STRING. This topic has great importance due to the clinical-oriented urgent need for discovering new biomarkers (either prognostic or predictive) in NSCLC patients. However, this study has several limitations. In summary, this paper needs major revision before it can be accepted.
2. Many sentences with grammatical errors are dispersed throughout the manuscript. It may benefit from an extensive editing by a native English speaking editor or a co-author.

Experimental design

The authors should download TCGA and GEO original data to analyze the expression, clinical significance of ITGA11, ITGB8 and ITGB4 in NSCLC, if not, the results are not solid.

Validity of the findings

1.Integrins family has many members, why authors just concentrate on studying these three members? Author should discuss the reasons, and also need to add the research progress of known prognostic biomarkers of NSCLC in the introduction “Ann Surg Oncol. 2017 Dec;24(13):4017-4024; Mol Biol Rep. 2012 Jan;39(1):335-41; Sci Rep. 2019 Aug 23;9(1):12329; Oncol Lett. 2019 Aug;18(2):1723-1732; Genes (Basel). 2019 Jan 10;10(1) pii: E36. doi: 10.3390/genes10010036”
2.For the OS analysis, multivariate cox regression analysis should be performed based on TCGA data, rather than using GEPIA only. Moreover, it is better to further validate the prognostic value of ITGB4 in tissue microarray with patients’ survival information by immunohistochemistry (IHC).

Additional comments

1.All images with low resolution are roughly made by online tools, and the quality of these figures should be improved.

Reviewer 2 ·

Basic reporting

1. There are some errors in grammar. Check whether the singular and plural forms of nouns are used correctly. For example, in the conclusion section of the abstract, “biomarker” should be written as “biomarkers”. “was” should be written as “were” in line 68.
2. The horizontal axis of figure 3A-3C is not clear.
3. The format of genes should be in italics throughout the manuscript.

Experimental design

1. Using bioinformatics analysis, the authors analyzed the expression of "integrin family ITGA11, ITGB4 and ITGB8" in human non-small cell lung cancer and its role in the prognosis of non-small cell lung cancer. Integrins are transmembrane proteins that mediate cell-cell and cell-extracellular matrix interactions. The authors should provide basic validation of ITGA11, ITGB4 and ITGB8 at the protein level. For example, is the ITGB4 protein correlated with its genetic level in NSCLC?

Validity of the findings

1. The author discussed the functions of ITGA1, ITGA4 and ITGA8 respectively. Is there any relationship among them?

Reviewer 3 ·

Basic reporting

In this manuscript, the authors investigated the expression levels of ITGA11, ITGB4, and ITGB8 in public cancer databases and found their applications as diagnostic or prognostic biomarkers. The results are clear and reproducible, therefore their findings seem to be useful for cancer research professionals. There are several points that need to be clarified:

1. The methods are basically a series of database searches. And there are no in-house experimental validations. Thus, the originality of their approaches may be weak.

Experimental design

2. Why were these three integrins selected? There are 18 subunits of integrins as the authors mentioned. And omics datasets make it possible to ideally investigate all genes.

Validity of the findings

3. Figure 2B: ITGB4 showed lower expression distribution on stage IV. ITGB4 seems good diagnostic and prognostic biomarkers for NSCLC (Figure 2A) and LUAD (Figure 4A), respectively. So, it seems contradictory that the higher tumor stage group has a lower expression profile of ITGB4.
4. The font size of Figure 5A is too small. It is difficult to understand which genes are co-expressed with three integrins.
5. Is there any reason to change datasets for ITGA11 (Hou et al. 2010), ITGB4 (Build et al. 2006), ITGB8 (Yamagata et al. 2003)? (l.156-l.164, Figure 5A) If there are three datasets, I would like to know concordance or discrepancy of co-expressed genes among them.

Additional comments

6. Although several omics platforms are used to investigate cancer genome, traditional immunohistochemistry (IHC) approaches are still routinely used for cancer diagnosis. Therefore, it is recommended to check protein expression levels of ITGA11, ITGB4, ITGB8 as diagnostic and prognostic biomarkers. (optional comments)

---

## Round 0.2 · Minor Revisions

Your revised manuscript has been reviewed by two of the three original reviewers. As you can see from their comments below, one of them is now satisfied with the revision while the other still raises additional minor points. Please re-revise the manuscript after carefully reading the comment. Thanks for your patience, in advance.

Reviewer 2 ·

Basic reporting

1. line 133, spaces before and after "<".
2. There are still some grammar mistakes, e.g.,line 145 and line 150, "are" should be "were".
3. In Figure 3, use the arrows to indicate where ITGA11, ITGB4, and ITGB8 are increased.
4. In line 174, delete "." after the word "results". Please check the punctuation in the manuscript.
5. Please explain what group the dotted line in Figure 5 represents

Experimental design

1. The author analyzed the mRNA expression level of ITGA11 and ITGB8 using GEO database. Has the author done q-PCR experiments to explore the mRNA expression of them in vitro?

Validity of the findings

no comments

Additional comments

no comments.

Reviewer 3 ·

Basic reporting

The authors have significantly improved the manuscript and have addressed all my comments. I have no additional comments.

Experimental design

no comment

Validity of the findings

no comment

Additional comments

no comment

---

## Round 0.3 · accepted · Accept

Now that all of the reviewers agree to accept your manuscript, I am happy to inform you that I will recommend its acceptance to the journal.

Reviewer 2 ·

Basic reporting

no comments

Experimental design

no comments

Validity of the findings

no comments

Additional comments

The manuscript could be accepted.